# Ketoanalogue Supplementation in Patients with Non-Dialysis Diabetic Kidney Disease: A Systematic Review and Meta-Analysis

**DOI:** 10.3390/nu14030441

**Published:** 2022-01-19

**Authors:** Vincenzo Bellizzi, Carlo Garofalo, Carmela Ferrara, Patrizia Calella

**Affiliations:** 1Division of Nephrology, University Hospital “San Giovanni di Dio e Ruggi d’Aragona”, 84131 Salerno, Italy; carmen17789@gmail.com; 2Division of Nephrology, University of Campania “Luigi Vanvitelli”, 80138 Naples, Italy; carlo.garofalo@unicampania.it; 3Department of Movement Sciences and Wellbeing, University “Parthenope”, 80133 Naples, Italy; patrizia.calella@assegnista.uniparthenope.it

**Keywords:** diabetic kidney disease, malnutrition, low-protein diet, very-low-protein diet, ketoanalogues, systematic review, meta-analysis, chronic kidney disease, CKD

## Abstract

The effects of supplemental ketoanalogues (KA) in patients with diabetic kidney disease (DKD) are not well characterized. Several databases for peer-reviewed articles were systematically searched to identify studies reporting outcomes associated with the effects of a low-protein diet (LPD) or very-low protein diet (VLPD) in combination with supplemental KA in adults with DKD. Meta-analyses were conducted when feasible. Of 213 identified articles, 11 could be included in the systematic review. Meta-analyses for renal outcomes (4 studies examining glomerular filtration rate; 5 studies examining 24-h urinary protein excretion), metabolic outcomes (5 studies examining serum urea; 7 studies examining blood glucose), clinical outcomes (6 studies examining blood pressure; 4 studies examining hemoglobin), and nutritional outcomes (3 studies examining serum albumin; 4 studies examining body weight) were all in favor of KA use in DKD patients. Data from individual studies that examined other related parameters also tended to show favorable effects from KA-supplemented LPD/VLPD. The regimens were safe and well tolerated, with no evidence of adverse effects on nutritional status. In conclusion, LPD/VLPD supplemented with KA could be considered effective and safe for patients with non-dialysis dependent DKD. Larger studies are warranted to confirm these observations.

## 1. Introduction

Diabetes mellitus is the leading cause of chronic kidney disease (CKD) worldwide, comprising approximately 30% of patients with non-dialysis CKD and 30% to 50% of patients reaching end stage renal disease (ESRD) [1,2,3]. Nutritional management with dietary protein restriction in addition to optimal glucose, lipid, proteinuria, and blood pressure (BP) control, is a central component of conservative treatment for patients with non-dialysis diabetic kidney disease (DKD) [4,5,6]. The primary goals of this intervention are to mitigate uremic symptoms, preserve or improve nutritional status, reduce the progression of CKD, and delay the requirement for dialysis therapy [4,5,7]. As DKD is characterized by increased systemic inflammation, insulin resistance, and hypercatabolism [2], protein-restricted diets might lead to essential amino acid (EAA) deficiency, low energy intake, and an increased risk of protein-energy wasting [2,8]. Current clinical practice guidelines for clinically stable, non-dialysis patients with DKD stages 3–5 are conflicting; the American Diabetic Association states that dietary protein intake should not be lower than 0.8 g/kg/day [9] because of the risk for malnutrition [10]; meanwhile, the National Kidney Foundation (NKF) suggests a low protein diet (LPD) providing dietary protein of 0.6–0.8 g/kg/day coupled with adequate energy intake (25–35 kcal/kg/day) to preserve renal function and maintain nutritional status [6].

Ketoacid analogues of EAA (ketoanalogues; KA) can serve as substrates for protein synthesis without the production of toxic nitrogenous waste products [11]. Thus, supplementation of a protein-restricted diet with KA allows a reduced intake of nitrogen while avoiding the deleterious consequences of inadequate dietary protein intake, mainly on muscle metabolism [12]. In nondiabetic patients with CKD, supplementation of a low protein diet (LPD) or very-low-protein diet (VLPD) with KA has been shown to have additional advantages on uremic control, reduce CKD-related cardiovascular risk factors, attenuate the disease progression, and delay the requirement for maintenance dialysis therapy while being nutritionally safe [13,14,15,16,17,18]. Indeed, the NKF guidelines strongly recommend KA-supplemented VLPD in metabolically stable nondiabetic CKD to improve outcomes [6]. The potential clinical benefits and harms of KA supplementation to protein-restricted diets in patients with non-dialysis DKD are less well characterized since the majority of the major trials did not include DKD patients, mostly because of the fear of malnutrition [19].

Thus, the primary aim of this review is to evaluate any potential benefits and harms associated with the use of KA in conjunction with a protein-restricted diet in non-dialysis DKD patients with regards to metabolic and nutritional outcomes and, secondarily, to explore its association with the progression of renal damage, ESRD, and mortality.

## 2. Materials and Methods

### 2.1. Search Strategy and Selection Criteria

A systematic literature review was performed in accordance with the guidelines of the Preferred Reporting Items for Systematic Reviews and Meta-Analyses (PRISMA) group [20]. MedLine^®^, Embase^TM^, the Cochrane Database of Systematic Reviews, the Cochrane Central Register of Controlled Trials, PubMed^®^, and Web of Science^TM^ were searched for articles on the use of KA in DKD patients published prior to 24 March 2021. The search terms are summarized in Table 1 (the search protocols are available in the Appendix A). Articles in English reporting outcomes from studies evaluating the effect of KA supplementation in patients with non-dialysis type 1 or 2 DKD were eligible for inclusion independent of study design, CKD stage, amount of dietary protein, or age. After the initial searches, duplicate articles were removed, then article titles and abstracts were screened to identify articles of interest. Editorials, narrative reviews, and abstracts without full-text articles were excluded. Afterwards, the full-text articles of the selected records were assessed according to prespecified criteria: (a) patients with type 1 or 2 DKD, (b) patients who had received KA, (c) at least one clinical outcome of interest was reported.

The reference lists of the full-text articles were also reviewed for relevant studies. Primary outcomes of interest included renal function parameters (glomerular filtration rate (GFR), serum creatinine, creatinine clearance, proteinuria), metabolic and clinical parameters (serum urea levels, fasting blood glucose, glycosylated hemoglobin, insulin dose, BP, hemoglobin, calcium, phosphate, potassium), and nutritional status (body weight, body mass index (BMI), serum albumin, cholesterol, triglycerides). Secondarily, ESRD/initiation of dialysis and safety issues (morbidity, mortality) were considered. Information related to study design, population, sample size, interventions, and reported outcomes was extracted and recorded on data extraction forms. For studies that did not meet the selection criteria, the reasons for exclusion were documented and are presented in Figure 1.

### 2.2. Quality Assessment

The Newcastle-Ottawa Scale (NOS) was used for quality assessment [21]. Risk of bias was assessed by considering 3 items: (1) selection of participants (containing 4 domains); (2) comparability (1 domain); and (3) outcome measure (3 domains). Each domain was rates as “Yes”, “No”, or “Unclear”. Each quality domain was categorized as low risk for bias (Yes) when the study reported adequate data and met the criteria, and high risk (No) when the study reported adequate data but did not meet the criteria for the specific quality domain. Studies not reporting data to allow quality assessment were categorized as “Unclear” and, therefore, potentially carrying a high risk of bias. “Yes” was scored 1 and “No” or “Unclear” were scored 0. Studies of high quality were defined as score ≥ 6 points; the overall quality assessment score is reported (Table 2). In addition, methodological quality assessments were performed using the Quality Assessment of Controlled Intervention Studies (for randomized studies), the Quality Assessment Tool for Observational Cohort and Cross-Sectional Studies (for nonrandomized studies), and the Quality Assessment Tool for Before-After (Pre-Post) Studies With No Control Group [22]. The results of these assessments are reported as a consensus rating for each domain of the relevant instrument for each study in the supplement material (Appendix A). Any disagreements in quality scores were resolved by discussion and consensus between 2 authors.

### 2.3. Statistical Analyses

Meta-analyses of unstandardized mean difference from baseline to end of treatment in the DKD on KA groups was performed for GFR, serum urea, urinary protein excretion, fasting blood glucose, hemoglobin, systolic blood pressure (SBP), diastolic blood pressure (DBP), serum albumin, and body weight considering a pre-post correlation of 0.5. When values in the studies were reported as median and interquartile range (IQR), mean and standard deviation (SD) were derived [31]. A conservative approach was used to pool results by using a random-effects model, which allowed for the variation of true effects across studies. Heterogeneity was analyzed using a chi-squared test on N-1 degrees of freedom and the I^2^ test with 95% confidence interval (CI) [32]. I^2^ values of 25%, 50%, and 75% corresponded to cutoff points for low, moderate, and high degrees of heterogeneity.

A sensitivity analysis was also conducted by omitting one study at a time and then reanalyzing the data to assess the change in effect size in order to exclude the possibility that a single study greatly affected heterogeneity [33]. In addition, univariate moderator analyses (female gender < or ≥40%; number of participants < or ≥50; follow-up < or ≥6 months) were performed to explore other possible sources of heterogeneity. Restricted maximum likelihood estimators were used to estimate model parameters [34]. Publication bias was assessed by funnel plot, Egger’s linear regression test [35], and Begg and Mazumdar rank correlation test [36]. Statistical significance was defined as a 95% CI with no overlap with the null effect value. A two-sided *p*-value < 0.05 was also considered significant. Analyses were performed using PROMETA 2 (INTERNOVI, Cesena, Italy), STATA/SE 11 (Stata Corporation, College Station, TX, USA), and R Studio version 1.1442.

## 3. Results

### 3.1. Literature Search and Study Characteristics

Using the search terms specified in Table 1, a total of 213 records were retrieved (Figure 1). After the removal of duplicate publications, 184 records were judged for eligibility based on title, abstract, and/or keywords. Of these, 25 publications underwent a full text review. A manual review of references cited in the studies yielded 2 additional citations, resulting in a total of 27 articles that were formally assessed for eligibility. Eleven of these studies were identified for the systematic review (Table 2) [2,8,12,23,24,25,26,27,28,29,30]. Agreement between 2 researchers was high. Of the 11 studies, 2 included patients with CKD stages 3 or 4 [12,23], 1 with CKD stage 4 [28], 2 with CKD stages 3–5 [2,8], 1 with CKD stages 4 or 5 [27], 2 with CKD stage 5 [26,29], and 3 did not report CKD stages, including patients with nephropathy [24,25,30]. The study periods spanned from 12 weeks to approximately 9 years. The amount of protein intake and KA was reported in several studies; protein intake ranged 0.25–0.8 g/kg/day, and EAA/KA prescription was different among studies, being reported as 3 × 600 mg/day, 144 mg/kg/day, 1 Tab/6 kg/day, 1 Tab/5–7 kg/day, 12 Tab/day, 1 Tab/10 kg/day, max 6 Tab/day, or 3 × 4 Tab each 0.63 g/day (Table 2). Dietary compliance was only reported in two studies, showing fewer patients were adherent to a VLPD (45% and 55%) [25,28], compared to those prescribed an LPD (69%) [25]. Eight studies, including a total of 402 analyzed patients, met the selection criteria for quantitative analysis and were used for meta-analyses [2,8,12,23,24,25,27,30]. Of these 8 studies, 3 were randomized controlled trials (RCT) in DKD, one comparing KA to placebo and to rhubarb [12], one comparing KA alone to KA plus *Nigella sativa* oil [23], and one comparing KA plus blood pressure medication to blood pressure medication alone [30]; 4 had a pre-post design comparing a previous period on either free diet [24,25] or LPD [8,27] with VLPD or LPD either plus KA, and 1 was a prospective, CKD-controlled, observational study on KA [2]. Furthermore, 3 of these 8 studies had <40% female participants [8,25,27] and 5 studies ≥40% female participants [2,12,23,24,30]; 3 studies had a follow-up of <6 months [12,23,30] and 5 studies had a follow-up of ≥6 months [2,8,24,25,27]. Of the remaining 3 studies not included in the meta-analyses, 2 were retrospective cohort studies using data from the National Health Insurance Research Database in Taiwan comparing KA users vs nonusers [26,29], and 1 was a subgroup analysis of KA-compliant vs. noncompliant patients [28]. 

The quality assessment scores obtained with the NOS are reported in Table 2. Most studies were susceptible to bias owing to factors ranging from incomplete reporting of statistical methods to inadequate power. Performance bias, detection bias, incomplete reporting, and small sample size were the most common sources of risk. With the NOS, a high study quality could be identified for 4 studies [2,12,23,30]. The results of the additional Quality Assessment Tools are reported in Appendix A.

### 3.2. Renal Outcomes

A summary of renal outcomes for the studies included in the systematic review is provided in Table 3. Five studies examined GFR as an outcome [8,12,23,27,28]; 4 of these studies could be included in the meta-analysis (pre-post comparison in DKD) [8,12,23,27]. The meta-analysis identified a nonsignificant unstandardized mean difference in GFR before and after KA use of 4.06 mL/min/1.73 m^2^ (95% CI: −1.84, 9.97; *p* = 0.177) (Figure 2). A significant heterogeneity was found among the studies (I^2^: 99.42%; *p* < 0.001). Percentage of females (9.98 mL/min/1.73 m^2^ [95% CI: 4.23, 15.73] for ≥40%; −1.44 mL/min/1.73 m^2^ [95% CI: −3.41, 0.53] for <40%; *p* < 0.001), duration of follow-up (9.98 mL/min/1.73 m^2^ [95% CI: 4.23, 15.73] for <6 months; −1.44 mL/min/1.73 m^2^ [95% CI: −3.41, 0.53] for ≥6 months; *p* < 0.001), and number of patients enrolled (9.98 mL/min/1.73 m^2^ [95% CI: 4.23, 15.73] for <50; −1.44 mL/min/1.73 m^2^ [95% CI: −3.41, 0.53] for ≥50; *p* < 0.001) were found to be significant moderators influencing the results of the meta-analysis. No publication bias was found as determined by funnel plot (Appendix A), linear regression test (*p* = 0.915), or rank correlation test (*p* = 0.999). Chang et al. observed that the decline in the GFR was significantly attenuated in both type of CKD patients, with and without diabetes, after switching from LPD alone to LPD + KA; such responsiveness was mainly related to diabetes [8]. In their multivariate analysis, diabetes and elevated serum albumin were identified as independent predictors of an attenuated GFR decline. Teodoru et al. reported that in an intermediate 1-year analysis, patients compliant to a VLPD supplemented with KA had a slightly slower, statistically nonsignificant GFR decline compared to noncompliant patients [28].

In 2 studies by Barsotti et al. with the same design but different follow-up durations in patients with DKD [24,25], the decline in creatinine clearance was reduced in patients who followed either an LPD or a VLPD with supplemental EAA/KA as compared to the previous unrestricted diet period. In addition, 2 RCTs that examined serum creatinine levels alone [12,23] found a significant improvement in DKD patients on KA compared to those who received placebo (Table 3).

Five studies that examined 24-h urine protein excretion found a significant decrease in patients treated with KA [12,23,25,27,30], with an unstandardized mean difference of −1.41 g/day (95% CI: −2.74, −0.08; *p* = 0.037) (Figure 2). Heterogeneity between studies was very high (I^2^: 99.42%; *p* < 0.001). Percentage of females (−0.72 g/day [95% CI: −1.64, 0.2] for ≥40%; −2.5 g/day [95% CI: −3.97, −1.03] for <40%; *p* = 0.044) and duration of follow-up (−2.5 g/day [95% CI: −3.97, −1.03] for ≥6 months; −0.72 g/day [95% CI: −1.64, 0.2] for <6 months; *p* = 0.044) were found to be significant moderators of this meta-analysis. No publication bias was found via funnel plot (Appendix A), linear regression test (*p* = 0.114), or rank correlation test (*p* = 0.602). Khan et al. and Zhu et al. found that DKD patients who received KA had significantly greater decreases in urine protein excretion compared to controls on placebo [12] or blood pressure medication alone [30].

### 3.3. Metabolic and Clinical Outcomes

A significant decrease in serum urea was observed for KA users in all 5 studies that examined this parameter [2,8,12,23,24], with an unstandardized mean difference of −29.61 mg/dL (95% CI: −54.39, −4.83; *p* = 0.019) (Figure 3). High heterogeneity was found (I^2^: 98.17%; *p* < 0.001), but sensitivity and moderator analyses did not explain this value. No publication bias was found as testified by the funnel plot (Appendix A), linear regression test (*p* = 0.236), and rank correlation test (*p* = 0.624). When comparing diabetic and nondiabetic patients, Bellizzi et al. [2] observed significant decreases in both patient groups at 6 months and at the final observation (Table 4); however, diabetic patients had higher serum urea levels at baseline and at follow-up compared to nondiabetic patients.

Seven studies [2,8,12,23,24,25,30] examined changes in blood glucose (Table 4); 6 of these studies were included in the meta-analysis [2,8,12,23,24,25]. Fasting blood glucose significantly reduced during KA use, with an unstandardized mean difference of −27.57 mg/dL (95% CI: −39.20, −15.94; *p* < 0.01) (Figure 3). High heterogeneity was found among studies (I^2^: 96.7%; *p* < 0.001). In the sensitivity analysis, no study exerted a significant effect. In moderator analyses, no role was found for female gender, duration of follow-up, or number of participants. No publication bias was found via funnel plot (Appendix A), linear regression test (*p* = 0.279), or rank correlation test (*p* = 0.652). Alam et al. [23] and Khan et al. [12] also observed a reduction in postprandial glucose levels after the intervention (Table 4). The studies by Barsotti et al. were the only ones that examined insulin requirements and found, corresponding, significantly decreased insulin needs [24,25]. The RCT by Zhu et al. [30] and the pre-post study by Mihalache et al. [27] found a significant decrease in glycosylated hemoglobin levels in DKD patients who received KA.

Changes in BP were assessed in 7 studies [2,8,12,23,27,30] (Table 4), 6 of which were included in the meta-analysis. A significant reduction in SBP was found, with an unstandardized mean difference of −10.01 mmHg (95% CI −14.86, −5.15; *p* < 0.001) (Figure 3). Heterogeneity was high (I^2^: 90.83%; *p* < 0.001). The change in DBP was not significant, with an unstandardized mean difference of −5.48 mmHg (95% CI: −15.13, 4.17; *p* = 0.266) and a high heterogeneity among studies (I^2^: 98.94%; *p* < 0.001). In the sensitivity analysis, no study exerted a significant effect for either SBP or DBP. No role for duration of follow-up, number of participants, or prevalence of female gender was found. No publication bias for SBP or DBP was found via funnel plot (Appendix A), linear regression test (*p* = 0.684 for SBP; *p* = 0.980 for DBP), or rank correlation test (*p* = 0.573 for SBP; *p* = 0.851 for DBP). Barsotti et al. [25] found a nonsignificant change in BP post intervention (no values were published). Khan et al. [12] and Zhu et al. [30] found a greater reduction in BP in DKD patients who received supplemental KA compared to DKD patients on placebo or controls who did not receive KA (Table 4).

Hemoglobin levels were assessed in 4 studies [2,8,12,23], and showed a nonsignificant change after KA use, with an unstandardized mean difference of 0.46 g/dL (95% CI: −0.41, 1.33; *p* = 0.302) (Figure 3). High heterogeneity was found (I^2^: 96.51%; *p* < 0.001). Duration of follow-up (−0.30 g/dL [95% CI: −0.48, −0.12] for ≥6 months; 1.27 g/dL [95% CI: 0.99, 1.55] for <6 months; *p* < 0.001) and number of patients enrolled (−0.30 g/dL [95% CI: −0.48, −0.12] for ≥50 patients; 1.27 g/dL [95% CI: 0.99, 1.55] for <50 patients; *p* < 0.001) were found to be significant moderators influencing the results of the meta-analysis. No publication bias was found via funnel plot (Appendix A), linear regression test (*p* = 0.271), or rank correlation test (*p*= 0.174). Bellizzi et al. [2] observed no differences in hemoglobin levels between DKD and CKD patients at 6 months follow-up (Table 4).

Micronutrients were rarely assessed during the KA intervention (Table 4). Calcium levels were evaluated by Bellizzi et al. [2], with no differences between pre-post intervention in DKD and as compared to CKD controls; Teodoru et al. [28] found no difference between DKD patients compliant or not to KA. Consistent data on changes in phosphate levels were reported by Bellizzi et al. [2]; they found a decline of phosphate which was significant only in nondiabetics after 6 months, while on the contrary it was significant in DKD after 2 years. The same study [2] was also the only one to evaluated serum potassium, which slightly, non-significantly reduced in both DKD and CKD patients after 6 months.

### 3.4. Nutritional Outcomes

Mean serum albumin levels were assessed in 3 studies (Table 5) [2,8,25], and significantly higher levels were found in DKD patients after a protein-restricted diet with supplemental KA compared to pre-intervention levels, with an unstandardized mean difference of 0.11 g/L (95% CI: 0.04, 0.19; *p* = 0.003) (Figure 4). No heterogeneity was found betweeen studies (I^2^: 0.00%; *p* = 0.848). No publication bias was found as testified by funnel plot (Appendix A), linear regression test (*p* = 0.211), and rank correlation test (*p* = 0.117).

Change in body weight was assessed in 5 studies (Table 5) [2,8,24,25,28], of which 4 [2,8,24,25] could be included in the meta-analysis. No significant difference in body weight was found, with an unstandardized mean difference of −1.32 kg (95% CI: −3.61, 0.98; *p* = 0.262) (Figure 4). Low heterogeneity was found (I^2^: 53.90%; *p* = 0.089). No publication bias was found via funnel plot (Appendix A), linear regression test (*p* = 0.247), or rank correlation test (*p* = 0.497). Teoduru et al. [28] found no significant difference between KA compliant or noncompliant DKD patients for changes in body weight. Mihalache et al. [27] found a small significant decrease in BMI after intervention (median: −1.5; 95% CI: −2.2, −0.83). Bellizzi et al. [2], which had the longest follow-up, found that body weight initially declined after the start of KA and thereafter remained unchanged for 3 years. In addition, this was the only study that provided a deeper insight into body composition and muscle function; no evidence of new onset protein-energy wasting or dynapenia was observed in the long-term, according to a multicompartment evaluation and dynamometry, respectively, in patients on a protein-restricted diet supplemented with KA, without any differences between CKD and DKD.

Lipids were rarely assessed during the KA intervention (Table 5). Cholesterol was assessed by Bellizzi et al. [2], and reduced significantly in DKD but not in CKD patients. Triglyceride levels during KA did not change in DKD [2], and with respect to either CKD controls [2] or non-compliant DKD patients [28].

### 3.5. Safety Outcomes

Minor adverse events related to KA were reported in 3 RCTs [12,23,30]; however, there was no noticeable difference when compared to controls on placebo or rhubarb [12], on KA plus *Nigella sativa* oil [23], or on blood pressure medication alone [30]. The most frequently observed adverse event in the study by Khan et al. [12] was nausea (2 patients in each group). Alam et al. [23] reported 3 mild to moderate adverse events in the KA group (1 nausea, 1 excessive thirst, 1 rashes). Zhu et al. [30] reported mild cases of gastrointestinal discomfort (2 patients in each group), transient ‘liver function damage’ (3 patients in KA group; 2 patients in control group), elevated serum creatinine (1 patient in KA group), and leukopenia (2 patients in KA group; 1 patient in control group).

In the cohort study by Wang et al. [29], no significant difference in mortality risk or in the combined outcomes of death or long-term dialysis was found between diabetic KA users compared to diabetic KA nonusers. The risk of long-term dialysis, however, was significantly greater in diabetic KA users compared to diabetic KA non-users. In contrast, the DKD cohort study by Chen et al. [26] saw a lower 5-year mortality rate in KA users compared to nonusers (34.7% vs. 42.7%), with an adjusted lower risk of mortality by 27% (HR: 0.73; 95% CI: 0.66, 0.82). In this study, the greatest risk reduction was observed in older patients (>70 y). This study also found a lower adjusted occurrence of major cardiovascular events in diabetic patients with stage 5 CKD who used KA versus KA nonusers.

## 4. Discussion

This systematic review and meta-analyses evaluated the effects of ketoanalogues in association with a protein-restricted diet in non-dialysis CKD patients with diabetes. The scientific literature focused on KA use in CKD is wide and detailed, while the studies that also or exclusively include diabetic subjects are still few, many of which have been conducted recently. Hence, the first step is to summarize the evidence in this particular group of subjects in order to clarify the efficacy, impact on renal damage and safety of this type of treatment. Overall, this review provided evidence of positive effects of protein-restricted diet combined with KA supplementation on renal, metabolic and clinical outcomes in DKD. Specifically, a protein-restricted diet plus KA was shown to improve the uremic burden [2,12,23,24] and attenuate CKD progression by preserving the decline in GFR [2,8,12,23,24,25,28], reducing the urinary protein excretion [12,23,30] and improving glucose control [2,12,23,25,30]. When compared to nondiabetic CKD patients, outcomes and safety in DKD patients were similar [2,8].

Based on strong evidence, the current NKF guidelines for nondiabetic non-dialysis patients with CKD stages 3–5 recommend an LPD with 0.55–0.60 g protein/kg/day or a VLPD with 0.28–0.43 g protein/kg/day supplemented with KA/amino acid analogues to meet protein requirements [6]. For patients with DKD stages 3–5, these guidelines suggest an LPD (0.6–0.8 g/kg/day) coupled with adequate energy intake to maintain stable nutritional status and optimize glycemic control; however, this statement is opinion rather than evidence-based and does not provide any recommendations on KA supplementation because of a lack of evidence. Of note, for overweight/obese diabetic patients, diabetologists recommend a low-energy, low-carbohydrate, high-protein diet, with the aim to prevent the clinical complications of diabetes and reduce cardiovascular risk [9]. In these guidelines it is stated that, when CKD is present, the dietary protein intake should not be <0.8 g/kg/day. This statement is also opinion and mainly founded on the fear of theoretical drawbacks such as impairment of glucose control and malnutrition associated with a protein-restricted diet [10]. This present systematic review with meta-analyses sheds some light on this controversy and provides evidence on the impact of restricted-protein diets on diabetes control and nutrition in DKD patients.

In non-dialysis CKD patients without diabetes, the reduced intake of protein delays ESRD due to the combined effects of slowing GFR decline and better controlling the metabolic and cardiovascular complication of CKD [37,38]. This effect is mostly associated with the high adherence to the diet [38]. In addition, a recent meta-analysis showed that KA-supplemented VLPD or LPD favorably attenuated GFR decline in CKD patients compared to an LPD alone [39]. In a further meta-analysis, a dietary protein intake <0.8 g protein/kg/day was found to have significant benefits on both GFR and proteinuria in early DKD [40]. Unfortunately, all the major RCTs on dietary protein intervention in advanced CKD do not include diabetic patients [19]. This review provides some evidence that the decline in GFR may be attenuated in DKD patients who receive a KA-supplemented protein-restricted diet, as the meta-analysis showed a slight, nonsignificant increase in GFR. The greatest reduction of GFR decline was reported in patients who were compliant with a KA-supplemented VLPD [28]. Although the impact of LPD plus KA on renal function in DKD patients cannot be conclusive due to the paucity of studies, our results are in agreement with data from the two major RCT in nondiabetic CKD patients, which showed a slower nonsignificant [41] or statistically significant [38] decline in GFR, depending on the characteristics and dietary adherence of the studied population. Our meta-analysis also found a significantly greater decline in proteinuria in DKD patients with the use of KA, even as compared to controls, indicating a potential additional benefit on renal damage when providing KA.

Furthermore, the control of BP and anemia is a major issue in DKD treatment to slow the progression of renal damage. Our meta-analyses found positive effects of KA-supplementation on reducing BP, though only the decline in SBP was statistically significant. The same reduction was observed when compared to controls without KA, and the reduction in BP was even greater when KA was given with BP medication compared to controls who received BP medication alone, suggesting that KA use may have an additional positive effect. These data confirm what was previously shown in nondiabetic CKD patients [42]. No change in hemoglobin was observed in any comparison; however, no data on erythropoietin use have been reported.

To compensate for the lower amount of energy coming from proteins, LPD usually contain more carbohydrates, which may worsen diabetic status. All the studies included in this review analyzing the glucose control, showed a statistically significant decline in fasting serum glucose levels with KA-supplemented LPD compared to LPD alone. In addition, declines in glycosylated hemoglobin and insulin requirements were reported in some studies, providing evidence of an overall improvement in glucose metabolism. It is unclear if these beneficial effects result from a reduction of carbohydrates in the diet with KA supplementation, or from general improvements of insulin resistance due to the better control of uremia as previously described for non-dialyzed uremic patients on an LPD [43].

Protein-energy wasting is a common concern in patients with diabetic CKD receiving LPD or VLPD, as the dietary protein intake may not meet the increased demand associated with the inflammation and catabolic state of the DKD patient, thus contributing to the loss of muscle mass [2,8,44]. In the studies included in the present review, the supplementation of LPD or VLPD with KA was seen to have no adverse effect on serum albumin and body weight, suggesting this approach could be nutritionally safe [2,8,24,25]. In DKD patients on LPD plus KA, serum albumin significantly increased in all studies, and body weight or BMI slightly decreased (not significantly) at the beginning of the diet, similar to the nondiabetic CKD controls on KA [2], and as previously shown in a large RCT in nondiabetic CKD [45]. When compared to KA-noncompliant DKD patients, no differences in body weight changes were observed [28]. These data confirm the same trends observed in body weight after starting an LPD in nondiabetic CKD [45]. Nonetheless, albumin and weight alone are not enough to reveal malnutrition, since only the evaluation of lean and fat masses, for instance, could detect sarcopenic malnutrition. One study evaluating body composition and muscle function in DKD and CKD over time [2] did not find any protein–energy wasting or impairment of muscle strength while on LPD plus KA, with no differences noted between CKD and DKD patients. In agreement, a recent meta-analysis revealed no higher risk of malnutrition for CKD patients receiving a KA-supplemented VLPD or LPD [39]. Overall, these data did not detect overt risk of protein–energy wasting in DKD while on KA.

Though we show positive effects on the progression of renal damage and major related risk factors, hard data on renal and patient outcomes are few, in part contradictory, and do not allow for any conclusions. Based on administrative data in the same population, Wang at al found higher risk of ESRD in DKD patients on KA [29], while Chen et al. reported a slightly lower incidence of ESRD among diabetics on KA [26]. These findings are partially in contrast to an RCT in nondiabetic CKD where ESRD improved under VLPD + KA [38], and, therefore, the issue still needs to be adequately addressed. Regarding mortality, Wang at al reported no difference between DKD patients who used KA and those who did not [29], and Chen et al. showed a significantly lower all-cause mortality in KA users compared to non-users [26]. Whether LPD + KA in DKD patients does impact on mortality or not, as found in nondiabetic CKD [19,46], still needs to be explored. Overall, these differences were possibly due to major selection bias and highlight the need for RCT.

Overall, the results from this systematic review and meta-analyses provide evidence that could be relevant for the management of CKD in diabetic patients. Nevertheless, our findings should be interpreted with caution in the context of some limitations. There are only a few studies, heterogeneity is high, and quality is rather low. Not all studies reported data regarding dietary protein intake, thereby limiting assessment of the nutritional efficacy of the prescribed regimens. Additionally, the use of different definitions for LPD and VLPD and the different doses of KA administered make it difficult to compare results between studies. This heterogeneity was, indeed, observed in the meta-analyses. Finally, consistent with available evidence, dietary compliance, which is low in CKD patients overall [47], was reported to be lower in patients receiving a VLPD [25,28]. This review cannot provide comprehensive data on the adherence to LPD + KA in DKD patients; however, the meta-analyses showed a statistically significant reduction in serum urea, indicating good adherence to, and the effectiveness of, the protein-restricted diets.

## 5. Conclusions

In conclusion, the summary of evidence provided by this systematic review evaluating a protein-restricted diet in combination with supplemental KA in patients with non-dialysis DKD suggests favorable effects on uremic burden, glucose metabolism, and progression of renal damage. The diets supplemented with KA were safe and well tolerated over periods ranging from 12 weeks to 9 years, with no evidence of adverse effects on nutritional status. Data on the initiation of dialysis and death were very few, and were likely encumbered by major bias, and cannot be conclusive. Based on these findings, LPD/VLPD supplemented with KA could be considered effective and safe for diabetic patients with advanced CKD. Adequately powered clinical studies are needed to confirm these observations.

## Figures and Tables

**Figure 1 nutrients-14-00441-f001:**
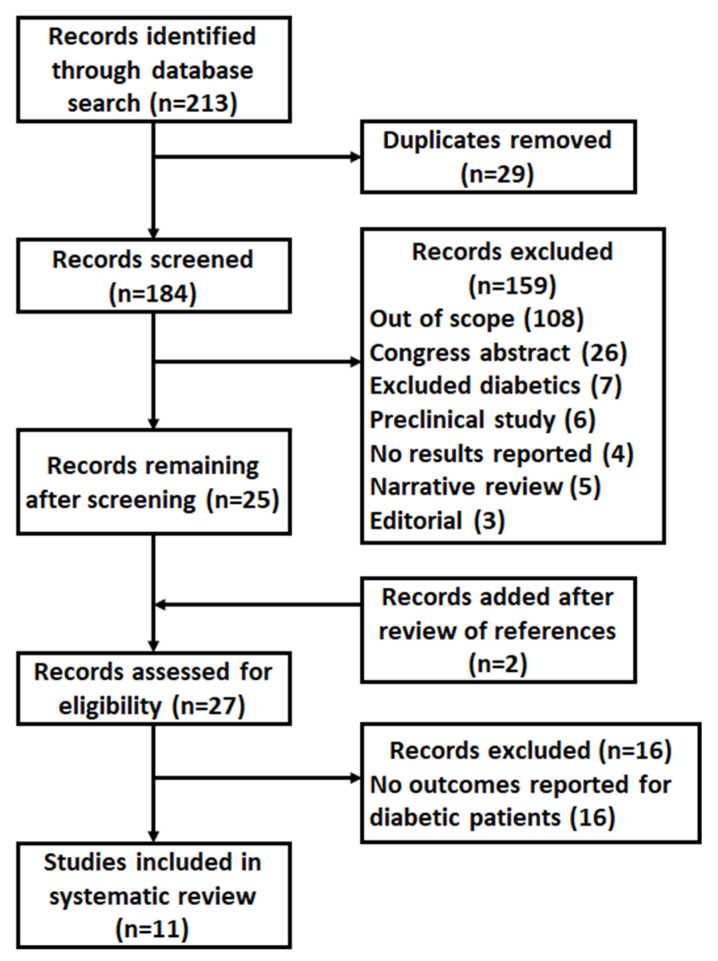
Preferred Reporting Items for Systematic Reviews and Meta-Analyses (PRISMA) study flow diagram.

**Figure 2 nutrients-14-00441-f002:**
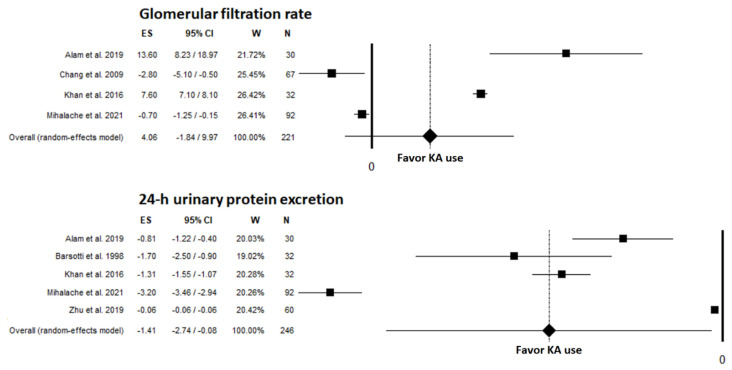
Meta-analyses for renal outcomes reported in studies included in the systematic review; GFR [8,12,23,27], 24-h upe [8,23,25,27,30]. KA, ketoanalogues.

**Figure 3 nutrients-14-00441-f003:**
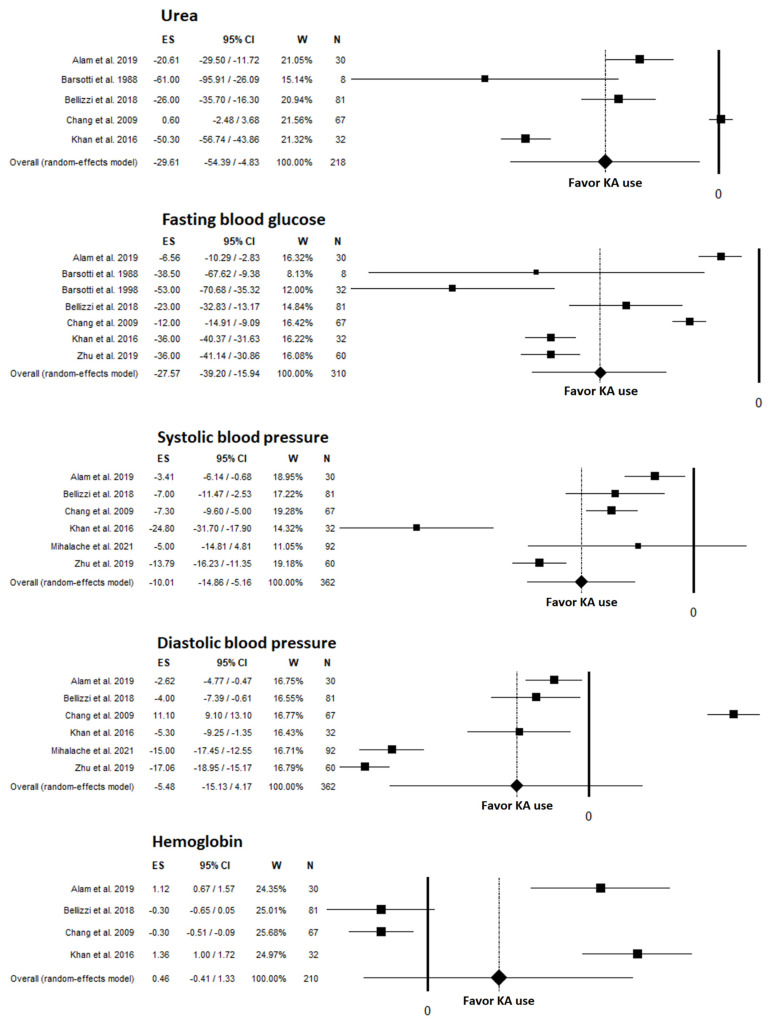
Meta-analyses for metabolic and clinical outcomes reported in studies included in the systematic review; Urea [2,8,12,23,24], FBG [2,8,12,23,24,25,30], SBP [2,8,12,23,27,30], DBP [2,8,12,23,27,30], Hb [2,8,12,23]. KA, ketoanalogues.

**Figure 4 nutrients-14-00441-f004:**
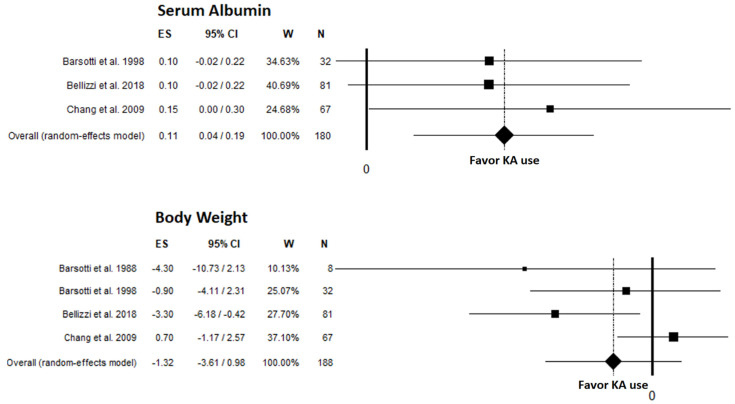
Meta-analyses for nutritional outcomes reported in studies included in the systematic review; Albumin [2,8,25], Weight [2,8,24,25]. KA, ketoanalogues.

**Table 1 nutrients-14-00441-t001:** Summary of search terms used.

Category	Search Terms
Population	Advanced kidney disease, advanced renal disease, chronic kidney disease, chronic kidney failure, chronic renal disease, chronic renal failure, chronic renal insufficiency, diabetes, diabetes mellitus, diabetic, diabetic kidney disease, diabetic nephropathy, pre-dialysis kidney disease, renal failure, renal insufficiency data
Intervention	Ketoacid, ketoacids, keto acid, keto acids, ketoacid supplements, ketoanalog, ketoanalogs, keto-analog, keto-analogs, ketoanalogue, ketoanalogues, keto-analogue, keto-analogues, KA, EAA

**Table 2 nutrients-14-00441-t002:** Characteristics of studies included in the systematic review.

Author Study Design	CKD Stage	Intervention Group			Control Group			Comparison (Control vs Patients)	Follow-Up (Months)	Quality Rating ^1^
		Disease, Patients (*n*)	Protein Intake (g/kg/day)	KA Pills	Disease, Patients (*n*)	Protein Intake (g/kg/day)	KA Pills			
Alam, 2019 RCT [23]	3/4	DKD, 32	NR	Ketolog^®^, 3 × 600 mg/day + oil	DKD, 30	NR	Ketolog^®^, 3 × 600 mg/day	KA vs KA + oil ^2^	3	6
Barsotti, 1988 Pre-post [24]	NR	Diabetic nephropathy, 8	VLPD/LPD (0.25–0.6)	EAA/KA 144 mg/kg/day	---	Free diet	None	No KA/diet vs. KA + LPD/VLPD	>12	5
Barsotti, 1998 Pre-post [25]	NR	Diabetic nephropathy, 32	VLDP/LPD (0.3/0.7) ^3^	Alfa Kappa^®^, 1 Tab/6 kg/day	---	Free diet	None	No KA/diet vs. KA + LPD/VLPD	>60	5
Bellizzi, 2018 Prospective observational [2]	3–5	DKD, 81 DKD, 29	LPD (0.5–0.6)	Ketosteril^®^, 1 Tab/5–7 kg/day	CKD, 116 CKD, 35	LPD (0.5–0.6)	Ketosteril^®^, 1 Tab/5–7 kg/day	CKD vs. DKD	6 & 36	8
Chang, 2009 Retrospective pre-post [8]	3–5	DKD, 67	LPD (≤0.6)	Ketosteril^®^, 12 Tab/day	--- CKD, 53	LPD (≤0.6)	NoneKetosteril^®^, 12 Tab/day	LPD vs. LPD + KA CKD vs. DKD	6	6
Chen, 2021 Retrospective cohort [26]	5	DKD, KA users ^4^ 1001	NR	NR	DKD, KA non-users 14,781	NR	None	KA non-users vs. KA users	60	---
Khan, 2016 RCT [12]	3/4	DKD, 32	NR	α-KA, 3 × 600 mg/day	DKD, 32 (placebo) DKD, 32 (rhubarb)	NR	None	Placebo vs. rhubarb capsules vs. KA ^5^	3	6
Mihalache, 2021 Pre-post(subgroup analysis) [27]	4/5	DKD, 92	LPD (0.6) ^6^	Ketosteril^®^, 1 Tab/10 kg/day	---	LPD (0.6)	None	LPD vs. LPD + KA	12	7
Teodoru, 2018 Prospective cohort (subgroup analysis) [28]	4	DKD, KA compliant 11	VLPD (0.3)	NR	DKD, KA noncompliant 9	LPD > 0.6 at least once	NR	KA noncompliant vs. KA compliant	12	---
Wang, 2020 Retrospective cohort [29]	5	DKD, KA user 67	LPD (0.6)	Ketosteril^®^ max. 6 Tab/day	DKD, KA non-user 107 CKD, KA non-user 58 CKD, KA user 98	LPD (0.6)	None None max. 6 Tab/day	KA non-user vs. KA user in DKD, and compared to CKD	up to 108 ^7^	---
Zhu, 2019 RCT [30]	NR	Diabetic nephropathy, 60	LPD advised	Crodi^®^, 3 × 4 Tab ea. 0.63 g/day	Diabetic nephropathy, 60	LPD advised	None	Irbesartan vs. irbesartan+ KA ^8^	3	6

CKD, chronic kidney disease; DKD, diabetic kidney disease; EAA, essential amino acids; KA, ketoanalogues; LPD, low-protein diet; NR, not reported; RCT, randomized controlled trial; Tab, tablet; VLPD, very-low-protein diet. ^1^ The Newcastle–Ottawa scale was used to assess the study quality, defining studies with a high quality with scores ≥ 6 points. ^2^ Both groups received conservative management incl. telmisartan, torsemide, iron, calcium, vitamin D3, erythropoietin, and insulin; *Nigella sativa* oil was given in doses of 2.5 mL/day. ^3^ VLDP with creatinine clearance between 6.5–19 mL/min, LPD with creatinine clearance between 22–20 mL/min. ^4^ Defined as patients who had ever used KA supplements before permanent dialysis. ^5^ All groups received conservative management including renal diet, telmisartan, and insulin. ^6^ Additional interventional components incl. intensive nutritional counselling + low salt intake (5 g/day) + hypertensive therapy. Protein diet was based on dry ideal body weight. ^7^ Duration of observation period was estimated based on results of Cox proportional hazard regression analysis. ^8^ Both groups received 0.15 g/day irbesartan, conventional management including diabetes diet, exercise, “blood sugar lowering to control blood sugar and blood pressure”, and patients were advised to pay attention to LPD.

**Table 3 nutrients-14-00441-t003:** Results from studies that examined renal parameters.

Parameter	Study	Outcome
GFR/estimated GFR	Alam et al., 2019 (RCT) [23]	Mean ± SD baseline vs week 12 (mL/min/1.73 m^2^): α-KA (control): 21.36 ± 6.66 vs 34.96 ± 17.17 ^1^ α-KA + *N. sativa* oil: 22.37 ± 6.85 vs 41.84 ± 17.92 ^1,2^
	Chang et al., 2009 (pre-post study) [8]	Mean ± SD pre vs post intervention (mL/min/1.73 m^2^): ^3^ Diabetics: 28.9 ± 10.2 vs. 26.1 ± 8.9 Mean ± SD change LPD v. LPD + KA (mL/min/1.73 m^2^/year): Diabetic: −12.3 ± 11.3 vs −4.0 ± 11.3 ^4^ Nondiabetic: −8.8 ± 8.7 vs −3.6 ± 6.4 ^4^ Independent predictors of attenuated decline with LPD + KA: Diabetes (HR 3.50; 95% CI 1.42, 8.62; *p* = 0.006) Elevated serum albumin (HR 3.13; 95% CI 1.30, 7.54; *p* = 0.011)
	Khan et al., 2016 (RCT) [12]	Mean ± SD baseline vs. 12 w (mL/min/1.73 m^2^): Placebo: 28.3 ± 1.53 vs. 32.4 ± 1.51 ^4^ Rhubarb: 28.4 ± 1.51 vs. 35.6 ± 1.46 ^1,5^ α-KA: 29.8 ± 1.48 vs. 37.4 ± 1.43 ^1,6^
	Mihalache et al., 2021 (pre-post study) [27]	Median (95% CI) pre vs. post intervention (mL/min/1.73 m^2^): 11.7 (11.2, 12.2) vs. 11.0 (10.3, 11.5) Median (95% CI) change pre to post intervention (mL/min/1.73 m^2^): −0.9 (−1.6, −0.1); *p* = 0.02
	Teodoru et al., 2018 (cohort study) [28]	Median (IQR) annual decline compliant (*n* = 11) vs. noncompliant ^7^ (*n* = 9) patients (mL/min/1.73 m^2^/year): From initial diagnosis of stage 4 CKD: 1.4 (2.02) vs. 4.1 (5.2); *p* = 0.006 From study baseline: 3.75 (7.4) vs. 4.1 (5.6); *p* = 0.37
Serum creatinine	Khan et al., 2016 (RCT) [12]	Mean ± SD baseline vs. week 12 (mg/dL): Placebo: 3.43 ± 1.14 vs 2.33 ± 0.87 ^1^ Rhubarb: 3.06 ± 1.38 vs. 1.82 ± 0.84 ^1,5^ α-KA: 3.68 ± 1.20 vs. 1.83 ± 0.70 ^1,6^
	Alam et al., 2019 (RCT) [23]	Mean ± SD baseline vs. week 12 (mg/dL): α-KA: 2.91 ± 0.74 vs. 2.07 ± 0.77 ^1^ α-KA + *N. sativa* oil: 2.85 ± 0.69 vs. 1.79 ± 0.67 ^1,2^
Creatinine clearance	Barsotti et al., 1988 (pre-post study) [24]	Mean ± SD change pre vs. post intervention (mL/min/mo): −1.38 ± 0.27 vs. −0.03 ± 0.37 ^1^
	Barsotti et al., 1998 (pre-post study) [25]	Mean ± SD change pre vs. post intervention (mL/min/mo): −0.9 ± 0.62 vs. −0.22 ± 0.21 ^1^
Urinary protein excretion	Alam et al., 2019 (RCT) [23]	Mean ± SD baseline vs. week 12 (g/day in 24-h urine): α-KA (control): 2.79 ± 1.19 vs. 1.98 ± 1.10 ^1^ α-KA + *N. sativa* oil: 2.68 ± 0.89 vs. 1.57 ± 0.84 ^1,6^
	Barsotti et al., 1998 (pre-post study) [25]	Mean ± SD pre vs. post intervention (g/day): 4.2 ± 2.6 vs. 2.5 ± 1.8 ^4^
	Khan et al., 2016 (RCT) [12]	Mean ± SD baseline vs. week 12 (g/day in 24-h urine): Placebo: 2.43 ± 1.24 vs. 1.43 ± 0.97 ^4^ Rhubarb: 2.38 ± 1.52 vs. 1.22 ± 0.78 ^1^ α-KA: 2.34 ± 0.78 vs. 1.03 ± 0.54 ^1,5^
	Mihalache et al., 2021 (pre-post study) [27]	Median (95% CI) change pre vs. post intervention (g/g creatininuria): −3.4 (−3.8, −3.1) ^1^
Urinary protein excretion	Zhu et al., 2019 (RCT) [30]	Mean ± SD baseline vs. week 12 (mg/day in 24-h urine): Irbesartan: 189.17 ± 12.67 vs. 150.46 ± 10.34 ^8^ Irbesartan + α-KA: 190.32 ± 10.57 vs. 129.67 ± 8.24 ^5,8^
		Mean ± SD baseline vs. week 12 (µg UAER/min in 24-h urine): Irbesartan: 102.28 ± 10.19 vs. 76.43 ± 8.57 ^8^ Irbesartan + α-KA: 104.46 ± 8.37 vs. 60.87 ± 6.68 ^5,8^
		Mean ± SD baseline vs. week 12 (g ALB/Cr per L in 24-h urine): Irbesartan: 178–47 ± 10.84 vs. 139.55 ± 10.46 ^8^ Irbesartan + α-KA: 175.17 ± 11.47 vs. 110.64 ± 12.29 ^5,8^

ALB/Cr, albumin/creatinine ratio; CI, confidence interval; CKD, chronic kidney disease; HR, hazards ratio; IQR, interquartile range; KA, ketoanalogues; LPD, low-protein diet; SD, standard deviation; UAER, urinary albumin excretion rate. Studies included in the meta-analysis for a given parameter are identified by bold font. ^1^ *p* < 0.001 (within group comparison). ^2^ *p* < 0.001 (versus control). ^3^ Values provided by the authors for this review. ^4^ *p* < 0.01 (within group comparison). ^5^ *p* < 0.05 (versus control). ^6^ *p* < 0.01 (versus control). ^7^ Noncompliance defined as any of the following: dietary protein intake > 0.6 g/kg/day, caloric intake < 30 kcal/kg/day, or ketoanalogues intake less than prescribed dose. ^8^ *p* < 0.05 (within group comparison).

**Table 4 nutrients-14-00441-t004:** Results from studies that examined metabolic and clinical parameters.

Parameter	Study	Outcome
Serum urea	Alam et al., 2019 (RCT) [23]	Mean ± SD baseline vs. week 12 (mg/dL): α-KA (control): 84.47 ± 26.11 vs. 63.86 ± 23.33 ^1^ α-KA + *N. sativa* oil: 82.85 ± 15.96 vs. 53.93 ± 14.35 ^1,2^
	Barsotti et al., 1988 (pre-post study) [24]	Mean ± SD pre vs. post intervention (mg/dL): 112.7 ± 57.7 vs. 51.7 ± 22.5 ^1^
	Bellizzi et al., 2018 (observational study) [2]	Mean ± SD baseline vs. 6 months (mg/dL): Diabetic: 131 ± 58 ^3^ vs. 105 ± 49 ^3,4^ Nondiabetic: 115 ± 52 vs 88 ± 36 ^4^ Mean ± SD baseline vs. final (duration 38 ± 13 mos, *n* = 64) (mg/dL): Diabetic: 130 ± 64 ^3^ vs. 108 ± 51 ^3^ Nondiabetic: 98 ± 48 vs. 90 ± 52
	Chang et al., 2009 (pre-post study) [8]	Mean ± SD pre vs. post intervention (mg/dL): ^5^ Diabetic: 47.8 ± 12.4 vs. 48.5 ± 13.2
	Khan et al., 2016 (RCT) [12]	Mean ± SD baseline vs. week 12 (mg/dL): Placebo: 97.3 ± 26.0 vs 59.5 ± 14.5 ^6^ Rhubarb: 99.0 ± 22.6 vs. 52.7 ± 16.3 ^1^ α-KA: 96.6 ± 21.4 vs. 46.3 ± 12.2 ^1,7^
Serum/blood glucose	Alam et al., 2019 (RCT) [23]	Mean ± SD baseline vs. week 12 (mg/dL)—fasting: α-KA: 98.76 ± 10.97 vs. 92.20 ± 9.76 ^1^ α-KA + *N. sativa* oil: 102.53 ± 21.26 vs. 94.25 ± 17.98 ^1^ Mean ± SD baseline vs. week 12 (mg/dL)—postprandial: α-KA: 124.09 ± 14.02 vs. 116.93 ± 10.77 ^1^ α-KA + *N. sativa* oil: 132.84 ± 30.08 vs. 122.12 ± 24.66 ^1^
	Barsotti et al., 1988 (pre-post study) [24]	Mean ± SD pre vs. post intervention (mg/dL)—fasting: 154.6 ± 46.4 vs. 116.1 ± 10.9; *p* = NS
	Barsotti et al., 1998 (pre-post study) [25]	Mean ± SD pre vs. post intervention (mg/dL)—fasting: 174 ± 58 vs. 121 ± 20 ^4^
	Bellizzi et al., 2018 (observational study) [2]	Mean ± SD baseline vs. 6 months (mg/dL)—fasting: Diabetic: 126 ± 52 ^3^ vs. 103 ± 29 ^3,4^ Nondiabetic: 97 ± 18 vs. 97 ± 25 Mean ± SD baseline vs. final (duration 38 ± 13 mos, *n* = 64) (mg/dL): Diabetic: 114 ± 55 vs. 104 ± 18 ^3^ Nondiabetic: 99 ± 16 vs. 94 ± 15
	Chang et al., 2009 (pre-post study) [8]	Mean ± SD pre vs. post intervention (mg/dL)—fasting: ^5^ Diabetics: 180.5 ± 12.5 vs. 168.5 ± 11.8
	Khan et al., 2016 (RCT) [12]	Mean ± SD baseline vs. week 12 (mg/dL)—fasting: Placebo: 140 ± 15.9 vs. 113 ± 14.3 ^1^ Rhubarb: 142 ± 15.6 vs. 108 ± 14.0 ^1,8^ α-KA: 141 ± 14.31 vs. 105 ± 9.5 ^1,7^ Mean ± SD baseline vs. week 12 (mg/dL)—postprandial: Placebo: 195 ± 31.2 vs. 167 ± 24.3 ^1^ Rhubarb: 192 ± 32.1 vs. 157 ± 16.6 ^1,8^ α-KA: 191 ± 35.2 vs. 153 ± 13.9 ^1,7^
	Zhu et al., 2019 (RCT) [30]	Mean ± SD baseline vs. week 12 (mg/dL ^9^)—fasting: Irbesartan: 153.2 ± 18.0 vs. 135.1 ± 18.0 ^4^ Irbesartan + α-KA: 153.2 ± 21.6 vs. 117.1 ± 18.0 ^4,8^
Glycosylated hemoglobin	Mihalache et al., 2021 (pre-post study) [27]	Median (95% CI) pre vs. post intervention (%): 8.5 (8.4, 8.7) vs. 8.1 (7.8, 8.3) Median (95% CI) change pre to post intervention (%): −0.5 (−0.3, −0.8) ^1^
	Zhu et al., 2019 (RCT) [30]	Mean ± SD baseline vs. week 12 (%): Irbesartan: 8.41 ± 0.61 vs. 8.20 ± 0.52 ^4^ Irbesartan + α-KA: 8.39 ± 0.53 vs. 7.76 ± 0.45 ^4,8^
Insulin requirement	Barsotti et al., 1988 (pre-post study) [24]	Mean ± SD pre vs. post intervention (IU/day): 51.4 ± 11.5 vs. 38.5 ± 3.0 ^1^
	Barsotti et al., 1998 (pre-post study) [25]	Mean ± SD pre vs. post intervention (IU/day): 49 ± 21 vs. 28 ± 10 ^6^
Blood pressure	Alam et al., 2019 (RCT) [23]	Mean ± SD baseline vs. week 12 (mmHg)—SBP: α-KA: 126.23 ± 8.32 vs. 122.82 ± 6.68 ^4^ α-KA + *N. sativa* oil: 132.5 ± 11.72 vs 130.62 ± 7.39 Mean ± SD baseline vs. week 12 (mmHg)—SBP: α-KA: 82.88 ± 6.88 vs. 80.26 ± 4.28 α-KA + *N. sativa* oil: 84.32 ± 11.66 vs. 82.46 ± 6.64
	Bellizzi et al., 2018 (observational study) [2]	Mean ± SD baseline vs. month 6 (mmHg)—SBP: Diabetic: 137 ± 20 ^3^ vs. 130 ± 21 Nondiabetic: 127 ± 15 vs 129 ± 16 Mean ± SD baseline vs. month 6 (mmHg)—DBP: Diabetic: 78 ± 18 vs. 76 ± 8 Nondiabetic: 76 ± 8 vs. 76 ± 10
	Chang et al., 2009 (pre-post study) [8]	Mean ± SD pre vs. post intervention (mmHg)—SBP: ^5^ Diabetics: 135.8 ± 10.2 vs. 128.5 ± 8.9 Mean ± SD pre vs. post intervention (mmHg)—DBP: ^5^ Diabetics: 72.5 ± 8.5 vs. 83.6 ± 8.2
	Khan et al., 2016 (RCT) [12]	Mean ± SD baseline vs. week 12 (mmHg)—SBP: Placebo: 151.3 ± 18.6 vs. 136 ± 16.2 ^6^ Rhubarb: 151.0 ± 21.6 vs. 132 ± 8.89 ^6^ α-KA: 153.8 ± 22.9 vs. 129 ± 9.53 ^1,8^ Mean ± SD baseline vs. week 12 (mmHg)—DBP: Placebo: 88.2 ± 10.4 vs. 85.98 ± 9.65 Rhubarb: 88.6 ± 11.4 vs. 84.63 ± 9.21 ^6^ α-KA: 89.0 ± 12.4 vs. 83.7 ± 10.0 ^1,8^
	Mihalache et al., 2021 (pre-post study) [27]	Median (95% CI) pre vs post intervention (mmHg): SBP: 125 (120, 140) vs. 120 (110, 130) DBP: 80 (75, 80) vs. 65 (60, 65) Median (95% CI) change pre to post intervention (mmHg): SBP: −5 (−15, 0): *p* = 0.08 DBP: −15 (−20, −10); *p* < 0.001
	Zhu et al., 2019 (RCT) [30]	Mean ± SD baseline vs. week 12 (mg/dL)—SBP: Irbesartan: 142.98 ± 10.01 vs. 132.56 ± 8.94 ^4^ Irbesartan + α-KA: 144.17 ± 10.32 vs. 130.38 ± 8.76 ^4^ Mean ± SD baseline vs. week 12 (mg/dL)—DBP: Irbesartan: 88.06 ± 7.96 vs. 72.48 ± 6.34 ^4^ Irbesartan + α-KA: 87.43 ± 8.01 vs. 70.37 ± 6.78 ^4^
Blood pressure	Barsotti, 1998 (pre-post study) [25]	No significant changes.
Hemoglobin	Alam et al., 2019 (RCT) [23]	Mean ± SD baseline vs. week 12 (g/dL): α-KA: 8.27 ± 1.34 vs. 9.39 ± 1.17 ^1^ α-KA + *N. sativa* oil: 8.95 ± 1.51 vs. 10.34 ± 1.18 ^1,2^
	Chang et al., 2009 (pre-post study) [8]	Mean ± SD pre vs. post intervention (g/dL): ^5^ Diabetics: 10.5 ± 0.68 vs. 10.2 ± 0.98
	Bellizzi et al., 2018 (observational study) [2]	Mean ± SD baseline vs. month 6 (g/dL): Diabetic: 12.0 ± 1.7 vs. 11.7 ± 1.5 Nondiabetic: 12.2 ± 2.2 vs. 12.3 ± 1.7
	Khan et al., 2016 (RCT) [12]	Mean ± SD baseline vs. week 12 (g/dL): Placebo: 8.81 ± 1.88 vs. 9.91 ± 1.58 ^1^ Rhubarb: 8.97 ± 2.03 vs. 10.0 ± 1.68 ^1^ α-KA: 8.94 ± 1.13 vs. 10.3 ± 0.89 ^1,8^
Calcium	Bellizzi et al., 2018 (observational study) [2]	Mean ± SD baseline vs. month 6 (mg/dL): Diabetic: 9.1 ± 0.8 vs. 9.1 ± 0.6 Nondiabetic: 9.3 ± 1.5 vs. 9.3 ± 0.6
	Teodoru, 2018 (cohort study) [28]	Median (IQR) change compliant (*n* = 11) vs. noncompliant ^10^ (*n* = 9) patients (mg/dL): from baseline to 12 months: −0.36 (0.61) vs. 0.07 (0.56); *p* = 0.1
Phosphate	Bellizzi et al., 2018 (observational study) [2]	Mean ± SD baseline vs. month 6 (mg/dL): Diabetic: 4.5 ± 1.3 vs. 4.1 ± 1.2 ^3,11^ Nondiabetic: 4.3 ± 1.0 vs. 3.7 ± 0.8 ^4^
		Mean ± SD baseline vs. year 2 (mg/dL): Diabetic: 4.3 ± 0.8 ^3^ vs. 3.8 ± 0.5 ^4^ Nondiabetic: 3.8 ± 0.8 vs. 3.6 ± 0.7
Potassium	Bellizzi et al., 2018 (observational study) [2]	Mean ± SD baseline vs. month 6 (mEq/dL): Diabetic: 5.0 ± 0.8 vs. 4.7 ± 0.6 Nondiabetic: 4.8 ± 0.7 vs. 4.7 ± 0.6

CI, confidence interval; DBP, diastolic blood pressure; KA, ketoanalogues; MAP, mean arterial pressure; RCT, randomized controlled trial; SBP, systolic blood pressure; SD, standard deviation. Studies included in the meta-analysis for a given parameter are identified by bold font. ^1^
*p* < 0.001 (within group comparison). ^2^
*p* < 0.001 (versus control). ^3^
*p* < 0.05 (versus nondiabetic). ^4^
*p* < 0.05 (within group comparison). ^5^ Values provided by the authors for this review. ^6^
*p* < 0.01 (within group comparison). ^7^
*p* < 0.01 (versus control). ^8^
*p* < 0.05 (versus control). ^9^ Original units of mmol/L were converted to mg/dL for consistency of presentation. ^10^ Noncompliance defined as any of the following: dietary protein intake > 0.6 g/kg/day, caloric intake < 30 kcal/kg/day, or ketoanalogues intake less than prescribed dose. ^11^
*p* = 0.06 (within group comparison).

**Table 5 nutrients-14-00441-t005:** Results from studies that examined nutritional parameters.

Parameter	Study	Outcome
Serum albumin	Barsotti et al., 1998 (pre-post study) [25]	Mean ± SD pre vs. post intervention (g/dL): 3.7 ± 0.4 vs. 3.8 ± 0.3; *p* = NS
	Bellizzi et al., 2018 (observational study) [2]	Mean ± SD baseline vs. 6 months (g/dL): Diabetic: 3.7 ± 0.6 ^1^ vs. 3.8 ± 0.4 Nondiabetic: 4.0 ± 0.6 vs 4.0 ± 0.4
	Chang et al., 2009 (pre-post study) [8]	Mean ± SD pre vs. post intervention (g/dL): ^2^ Diabetic: 3.2 ± 0.6 vs. 3.4 ± 0.7
Body weight	Barsotti et al., 1988 (pre-post study) [24]	Mean ± SD pre vs. post intervention (kg): 69.1 ± 9.9 vs. 64.8 ± 8.5; *p* = NS
	Barsotti et al., 1998 (pre-post study) [25]	Mean ± SD pre vs. post intervention (kg): 68.7 ± 10.1 vs. 67.8 ± 8.1; *p* = NS
Body weight	Bellizzi et al., 2018 (observational study) [2]	Mean ± SD baseline vs. 6 months (kg): ^2^ Diabetic: 68.5 ± 14.0 vs. 65.2 ± 12.3 ^3^ Nondiabetic: 66.0 ± 15.0 vs 63.9 ± 15.1 ^3^
	Chang et al., 2009 (pre-post study) [8]	Mean ± SD pre vs. post intervention (kg): ^2^ Diabetic: 64.5 ± 6.8 vs. 65.2 ± 8.5
	Teodoru, 2018 (cohort study) [28]	Median (IQR) change compliant (*n* = 11) vs. noncompliant ^4^ (*n* = 9) patients (kg): from baseline to 12 months: 1.75 (4.5) vs. 1.5 (7); *p* = 0.44
Cholesterol	Bellizzi et al., 2018 (observational study) [2]	Mean ± SD baseline vs. 6 months (mg/dL): Diabetic: 186 ± 50 vs. 165 ± 37 ^3^ Nondiabetic: 177 ± 57 vs 166 ± 39
Triglycerides	Bellizzi et al., 2018 (observational study) [2]	Mean ± SD baseline vs. 6 months (mg/dL): Diabetic: 183 ± 86 vs. 167 ± 83 Nondiabetic: 171 ± 88 vs 165 ± 78
	Teodoru, 2018 (cohort study) [28]	Median (IQR) change compliant (*n* = 11) vs. noncompliant ^4^ (*n* = 9) patients (mg/dL): from baseline to 12 months: 13.59 (45.34) vs. 73.79 (135.68); *p* = 0.07

NS, not significant; SD, standard deviation. Studies included in the meta-analysis for a given parameter are identified by bold font. ^1^
*p* < 0.05 (versus nondiabetic). ^2^ Values provided by the authors for this review. ^3^
*p* < 0.05 (within group comparison). ^4^ Noncompliance defined as any of the following: dietary protein intake > 0.6 g/kg/day, caloric intake < 30 kcal/kg/day, or ketoanalogues intake less than prescribed dose.

## Data Availability

The meta-analyses data can by provided by the corresponding author on request at the e-mail address: vincenzo@bellizzi.eu.

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
