# Peer review of "Ketoanalogue Supplementation in Patients with Non-Dialysis Diabetic Kidney Disease: A Systematic Review and Meta-Analysis"

_nutrients, 2022, doi:10.3390/nu14030441_

Round 1

Reviewer 1 Report

The authors conducted a systematic review and meta-analysis of the effects of ketoanalogue supplementation and a low protein/very low protein diet on diabetic kidney disease (DKD).
As the authors also discuss, protein-restricted diets have conflicting guidelines on their pros and cons, and no certain conclusion has been reached, but there is no doubt that it is an important debate. In this regard, we can conclude that the authors' review and meta-analysis are necessary.
Although the 11 papers reviewed include some that are rather old, I judge that there are no major problems.
The authors assessed renal function with GFR/eGFR, Cr/creatinine clearance, urinary protein, and serum urea, and diabetes with blood glucose, HbA1c, and insulin requirements, as well as blood pressure, Hb, and weight.
In order to increase the value of the paper, there is something I would really like to see added about the effects and evaluation of DKD and ketoanalogue Supplementation.

â‘ As the authors discuss in the introduction, lipid control is also important in the treatment of DKD. Lipids are easily affected by diet, and patients with renal failure often have lipid abnormalities as well. Therefore, it would be better to include cholesterol and other parameters in the evaluation.

â‘¡As for the evaluation of chronic renal failure, the author and his colleagues have described GFR/eGFR/BUN/Cr/Hb, but electrolytes should be evaluated in renal failure. In addition, if ketone bodies are taken into the body for patients with diabetes and renal failure, the acid-base balance of the blood should be considered. Please add to the evaluation and discussion.

Author Response

Dear Madame/Sir,

Thank you for your favourable comments on the paper.

Please, find below the point-by-point-reply.

  1. As the authors discuss in the introduction, lipid control is also important in the treatment of DKD. Lipids are easily affected by diet, and patients with renal failure often have lipid abnormalities as well. Therefore, it would be better to include cholesterol and other parameters in the evaluation.

Thank you for evidencing this important point. Unfortunately, only few information on lipids have been reported in most papers. Nonetheless, the available data on cholesterol and triglycerides have been now reported in the Method and Result sections and Table (Lines 88-89, pg. 3; Lines 365-368, pg, 16; Table 5, pg. 17).

  1. As for the evaluation of chronic renal failure, the author and his colleagues have described GFR/eGFR/BUN/Cr/Hb, but electrolytes should be evaluated in renal failure. In addition, if ketone bodies are taken into the body for patients with diabetes and renal failure, the acid-base balance of the blood should be considered. Please add to the evaluation and discussion.

Thank you again for underlining this major issue. Unfortunately, also for electrolytes limited information were reported and we added it to the Method and Result sections and Table (Lines 88-89, pg. 3; Lines 308-316, pg, 11; Table 4, pg. 14). On the contrary, data on the acid-base balance were reported in only one paper with very few patients and were not reported.

Reviewer 2 Report

This paper is a meta-analysis of studies that have utilised ketoanalogues in population with diabetic kidney disease. The paper is well written and conducted properly, with bias analyses and reported according to PRISMA guidelines.

There is one small error, I think I may have noticed.

In Table 2 – Alam 2019 says both the intervention and control group are with “+ oil” -> is this correct?

Author Response

Dear Madame/Sir,

Thank you for the favourable comments on the paper.

Please, find below the point-by-point-reply.

  1. There is one small error, I think I may have noticed. In Table 2 – Alam 2019 says both the intervention and control group are with “+ oil” -> is this correct?

Sorry for this mistake. Now it has been corrected in Table 2.